# A Review of IsomiRs in Colorectal Cancer

**DOI:** 10.3390/ncrna9030034

**Published:** 2023-06-07

**Authors:** Molly A. Lausten, Bruce M. Boman

**Affiliations:** 1Cawley Center for Translational Cancer Research, Helen F. Graham Cancer Center & Research Institute, Newark, DE 19713, USA; 2Department of Biological Sciences, University of Delaware, Newark, DE 19713, USA; 3Department of Pharmacology & Experimental Therapeutics, Thomas Jefferson University, Philadelphia, PA 19107, USA

**Keywords:** colorectal cancer, microRNAs, isomiRs

## Abstract

As advancements in sequencing technology rapidly continue to develop, a new classification of microRNAs has occurred with the discovery of isomiRs, which are relatively common microRNAs with sequence variations compared to their established template microRNAs. This review article seeks to compile all known information about isomiRs in colorectal cancer (CRC), which has not, to our knowledge, been gathered previously to any great extent. A brief overview is given of the history of microRNAs, their implications in colon cancer, the canonical pathway of biogenesis and isomiR classification. This is followed by a comprehensive review of the literature that is available on microRNA isoforms in CRC. The information on isomiRs presented herein shows that isomiRs hold great promise for translation into new diagnostics and therapeutics in clinical medicine.

## 1. Introduction

### 1.1. Initial Discovery of MicroRNAs

MicroRNAs (miRNAs) are short non-coding RNAs that negatively regulate gene expression and were first discovered in Caenorhabditis elegans in 1993 by both the Ruvkun and Ambros research groups in their examination of lin-4 [1,2]. They were originally believed to be a novel discovery in nematodes of little consequence, and it was not until 7 years later that let-7, the first miRNA confirmed in humans, sparked a growing interest in the field of miRNA research [3]. Today, it has been shown that miRNAs play a role in all model systems, from plants to animals as well as several viruses [4,5,6]. It is now widely believed that miRNAs regulate nearly half of all human genes [7]. They are an important piece of the puzzle when seeking to understand the vast network of gene regulation and dysregulation in the transcriptome.

### 1.2. First Implications of miRNAs in Cancer

The first implication for the role of miRNAs in cancer came shortly after the initial discovery in humans. Calin et al. discovered a region of 13q14, frequently deleted in patients with B-cell chronic lymphocytic leukemia, that was home to miRNA cluster 15 and 16 [8]. Continued research by this lab showed that a loss of miRNAs 15/16 led to an upregulation of the oncogene BCL-2, which functions to suppress apoptosis [9]. Since then, the field has continued to grow, with miRNAs being shown to play either a tumor-suppressive or an oncogenic role. Rigorous translational research is under way to try and characterize the role miRNAs play in cancer and how this information can be used to develop more effective treatments as well as to help with early detection of various cancers.

## 2. Colon Cancer and miRNAs

### 2.1. Brief Overview of Colon Cancer

Colorectal cancer (CRC) is the third most common cancer worldwide and has one of the highest mortality rates [10]. If diagnosed early, CRC has a relatively high 5-year survival rate. However, if the cancer has metastasized and is caught at a later stage, the survival rate will fall abysmally. This high mortality rate is concerning because of a trend of increased incidence of early-onset CRC in patients under the age of 45. This incidence of CRC in young patients has been increasing in the United States since the late 1990s [11]. This trend has led to a great emphasis on the importance of early diagnosis as well as continued research to find an effective cure for cancers diagnosed at advanced stages [12,13,14]. Dysregulation of miRNAs has been shown to play a role in all stages of CRC progression, from initiation to metastasis [15].

### 2.2. Early Implications of miRNAs in Colon Cancer

miRNAs are seen as having potential prognostic and therapeutic potential in colon cancer [16]. The first cancer that a miRNA pattern was characterized for was CRC. Michael et al. described an expression pattern of 28 miRNAs in colon cancer that showed differential expression in precancerous and neoplastic colon tissues [17]. Since then, many studies, which have previously been reviewed, have been completed on miRNA expression pattern changes in CRC [18,19]. Yang et al., with their release of DbDEMC 2.0 (a compilation of profiling data sets), which documents differentially expressed miRNAs, showed that colon cancer had the highest number of dysregulated miRNAs in cancer compared to normal tissue among the 36 cancer types reviewed [20]. These selected studies highlight the importance of understanding the role that miRNAs play in colon cancer as potential diagnostic biomarkers and also as a key piece of the puzzle to understand what happens mechanistically during colon cancer progression.

## 3. Canonical Biogenesis of miRNAs

### 3.1. Biogenesis in the Nucleus

miRNAs are one type of small, non-coding RNA that, when fully mature, range from only 18 to 24 bp in length. One of the ways they are distinguished from the variety of other small RNAs is their unique biogenesis [21,22]. This review will only detail the canonical process of biogenesis. For non-canonical biogenesis, the reader is referred to several reviews on the subject [23,24]. miRNAs are typically encoded in miRNA genes. However, encoding regions have great variety, and although they have been found primarily in introns, they can also be found in untranslated regions (UTRs) of protein-coding genes and in introns or exons of other non-coding RNAs [6]. miRNAs are primarily transcribed by RNA polymerase II. There is also evidence that RNA polymerase III can produce a primary miRNA through normal transcription processes [25,26]. A primary miRNA forms a hairpin structure, several kilobases in length, and is 5′ capped and possibly 3′ polyadenylated [24,27]. The primary miRNA is further processed within the nucleus by the microprocessor complex. Its main components are DiGeorge Syndrome Critical Region 8 (DGCR8) and the ribonuclease III enzyme DROSHA (Figure 1a). DROSHA cleavage is modulated by the RNA-binding protein (RBP) serine/arginine-rich splicing factor 3 (SRSF3), which recognizes a motif in primary miRNAs [28,29]. The primary miRNA hairpin structure is recognized and bound by two molecules of DGCR8, and DROSHA cleaves the primary miRNA duplex around 22 bps from the junction site, releasing a precursor miRNA 50–80 nucleotides in length [30,31,32]. Precursor miRNAs are exported from the nucleus by direct interaction with exportin 5 and a Ras-related nuclear protein–guanosine-5′-triphosphate-ase (RAN-GTP) complex [33]. See Figure 1b for a visual representation of the above section.

### 3.2. Biogenesis in the Cytoplasm

Once the miRNAs are in the cytoplasm, another RNAse enzyme III, DICER, recognizes the 3′ overhang that was generated by the DROSHA cleavage and cleaves the precursor miRNAs to generate a short double-stranded RNA fragment of around 20–25 nt in length [31,34,35]. To date, there is only one miRNA that, via its unique biogenesis, is able to bypass DICER, namely miR-451. It is highly abundant in erythrocytes and is an essential miRNA in red blood cell maturation [36]. This highlights the importance of DICER activity in generation of miRNAs. RNA binding proteins TAR RNA binding protein 2 (TARBP2) and kinase R-activating protein (PACT) associate with DICER to increase stability. This results in a mature miRNA duplex. This double-stranded miRNA is transferred onto an Agronaute (AGO) protein in a direct transfer from the DICER complex to a member of the AGO protein family (AGO1–4) [6]. This duplex is loaded into an AGO family of proteins in an ATP-dependent manner [37]. Typically, the strand with lower 5′ stability or 5′ uracil is more likely to be loaded into AGO and is called the guide strand [6]. The passenger strand is then unwound from the guide strand and subsequently degraded. The exact mechanism of how strand selection occurs has not been fully elucidated, but it is a subject of great research interest [38]. The strands are named based on which strand they are derived from within the precursor miRNA hairpin. The directionality of the miRNA strand determines the name of the mature miRNA form. For example, the 5p strand arises from the 5′ end of the precursor miRNA hairpin. The ratio of 5p- to 3p-strand miRNAs that are AGO-loaded has been found to vary greatly depending on cell type and environment [23,38]. See Figure 1b for a visual representation of the above section.

### 3.3. miRNA-Induced Silencing Complex Formation

The guide strand and the AGO form a complex known as the minimal miRNA-induced silencing complex (miRISC). Nucleotides 2–8 on the 5′ end are the seed region and used to target the mRNAs. The guide strand is used to target complementarity sequences of target mRNAs, known as miRNA response elements, characteristically found in 3′ UTR. Sequence complementarity determines if there is translational inhibition and target mRNA decay or AGO2 dependent-splicing based on full complementarity with the target sequence [23]. Table 1 lists accessory proteins that are involved in the biogenesis process but have not been included in the narrative above.

**Table 1 ncrna-09-00034-t001:** A table of selected accessory proteins involved in miRNA biogenesis.

Gene/Protein ID	Location	Function in Biogenesis	Citations
SMAD4	Nucleus	SMAD4, in complex with SMAD2, is involved in the TGFβ/BMP signaling pathway, and upon activation, they both are able to activate miRNA precursor transcription.	[39,40,41,42]
SMAD2	Nucleus	SMAD4, in complex with SMAD2, is involved in the TGFβ/BMP signaling pathway, and upon activation, they both are able to activate miRNA precursor transcription. SMAD2 also expedites DROSHA processing of primary miRNAs within the nucleus.	[39,42,43,44]
DDX17	Nucleus	DDX17 recruits DROSHA as well as playing a role in the binding of primary miRNAs by DROSHA.	[31,42,43]
SRSF3	Nucleus	SRSF3 binds to CNNC motifs and recruits DROSHA to the basal junction for processing of primary miRNAs.	[28,29]
PACT	Cytoplasm	PACT synchronizes precursor miRNA cleavage by DICER1 and contributes to miRISC assembly.	[42,45,46]
KHSRP	Cytoplasm	KHSRP promotes the maturation of a subset of miRNAs by binding at their terminal loop.	[42,47]
ADAR	Cytoplasm	Adenosine deaminase acting on RNA (ADAR) is a double-stranded RNA-specific enzyme that can act on primary and precursor miRNA stem loops to deaminate an adenosine to an inosine. This can inhibit precursor miRNA cleavage by DICER1.	[42,48,49]
LIN28A/B	Cytoplasm	LIN28A/B plays a role in inhibiting DICER1 cleavage by binding to the terminal loop of a subset of precursor miRNAs and recruiting ZCCHC11 or ZCCHC6 that leads to uridylation of precursor miRNAs.	[42,50,51,52]
ZCCHC11 (TUT4) and ZCCHC6 (TUT7)	Cytoplasm	Uridylates precursor miRNAs, leading to DICER1 processing inhibition.	[42,51,53]
TUDOR-SN	Cytoplasm	TUDOR-SN is a ribonuclease, specific to inosine, containing primary transcripts.	[54]
TNRC6A	Cytoplasm	TNRC6A interacts with AGO proteins and plays a role in mRNA degradation.	[42,55,56]
PARN	Cytoplasm	PARN is an (A)-specific ribonuclease that degrades miRNAs that have been adenylated by PAPD4 and PAPD5.	[57]
PAPD4 and PAPD5	Cytoplasm	PAPD4 and PAPD5 are nucleotidyl transferases, with adenosyltransferase activity, that lead to miRNA degradation.	[57,58]

Table idea was adapted from [42].

## 4. Discovery of miRNA Isoforms and Their Classifications

Advances in sequencing technology in recent years have uncovered another layer of complexity for miRNAs with the identification of isoforms of miRNAs, known as isomiRs. These isoforms can vary in length and sequence from their reference or template miRNAs. Multiple types of isomiRs exist, with (*i*) changes at the 5′ end, (*ii*) changes at the 3′ end, (*iii*) variations within the sequence with no changes in length or (*iv*) a mix of the types listed above. The isomiR classifications are shown visually in Figure 2, below.

## 5. Generation of Isoforms through the Biogenesis Pathway

### 5.1. DROSHA/DICER-Induced Inaccuracies

A subset of isomiRs is believed to be produced through the biogenesis process via specific variations in cleavage, mainly by DICER and DROSHA, in the canonical miRNA biogenesis pathway. There is evidence that structural features of primary miRNAs and precursor miRNAs can lead to cleavage variations and generation of 5′ and 3′ variations. A recent study showed that nearby G-U wobble pairs, as well as internal loops of the primary miRNA stem, can induce non-canonical cleavage by DROSHA, leading to generation of 5′ isomiRs. These G-U wobble pairs cause increased structural flexibility that then lead to non-canonical cleavage [59]. Other features that have been shown to affect processing include structural features such as stem loop length and nucleotide motifs, as well as the activity of accessory proteins. This review will not detail all elements that have been found; however, several reviews are available that discuss the structural elements that affect cleavage fidelity [60,61]. A recent study by Shkurnikov et al. attempted to quantitatively estimate the inaccuracies in DROSHA and DICER by analyzing matched pairs (*n* = 8) of normal and malignant colon tissue that were available in the cancer genome atlas (TCGA) database [62]. They used these samples to analyze the isomiR profiles of cancer vs. normal, and, based on the location of the sequence variation, they were attributed to either DROSHA or DICER inaccuracies. Overall, this showed that the cytoplasmic DICER had a higher percentage of errors compared to the nuclear DROSHA enzyme and that the cancerous versus normal tissues were significantly different in the paired samples [62]. Zhiyanov et al. expanded on this work with a pan-cancer bioinformatics analysis, looking at the main DROSHA/DICER cleavage positions in various tissues, to determine if isomiR generation is dependent on cancer type. Utilizing TCGA dataset, they isolated the top twenty percent of the most highly expressed isomiRs for analysis. They found that DROSHA/DICER cleavage sites were generally not tissue-specific, particularly at the 5′ end. A majority of isomiRs have variability at the 3′ end, which provides further evidence that DICER is more prone to errors than the nuclear DROSHA [63].

There is also evidence that other members of the biogenesis pathway can play a role in isomiR generation: in particular, the binding partners of DROSHA and DICER. SRSF3 is an RBP that binds to primary miRNAs at a motif downstream of the DROSHA cleavage site. Bofill-De Ros et al. investigated the impact that SRSF3 could have on alternative DROSHA cleavage [59]. They utilized the breast cancer dataset (BRCA) in TCGA database and compared the miRNA profiles of BRCA patients with high levels and low levels of SRSF3. They found that patients with higher levels of SRSF3 had higher levels of most miRNAs, and a subset of miRNAs also showed differences in the fidelity of the cleavage [59]. Initial studies showed that the DICER binding partners TRBP and PACT can modulate DICER activity and cleavage sites [45]. Further evidence showed that DICER binding partner TRBP can lead to generation of isomiRs that are one nucleotide longer than the canonical sequence. This was further confirmed by a TRBP gene knockout that showed a subset of precursor miRNAs with shifted seed sequences [64,65,66]. It is also important to note that changes in these RBPs’ expression levels would not only affect isomiR generation and expression but also affect other parts of RNA metabolism that could potentially confound and complicate research. For example, SRSF3 is involved in regulating alternative splicing of over 90% of protein coding genes [67].

The selected studies above, although not comprehensive, highlight an enduring theory in the field that a subset of isomiRs are generated through the biogenesis pathway by a multitude of mechanisms.

### 5.2. Biogenesis Mutations Characterized in Colon Cancers

There is also evidence that genetic and epigenetic alterations in proteins involved in different stages of miRNA biogenesis are altered in various cancers, including colon cancer, which is detailed by the studies mentioned below. Notably, the mutations in those genes are hypothesized to be another source of aberrant isomiR generation in cancer. To determine how miRNA biogenesis is dysregulated, a recent study analyzed whole-exome sequencing datasets of over 10,000 paired cancerous and normal samples and characterized cancerous somatic mutations in miRNA biogenesis genes [42]. Those researchers identified and characterized close to 4000 somatic mutations in 29 miRNA biogenesis genes and demonstrated that some genes are frequently mutated in specific cancers and some have persistent hotspot mutations. In their study, Galka Marciniak et al. analyzed 411 cases of colon cancer retrieved from TCGA database. TCGA is a tumor dataset comprising 10,271 primary cancer samples, including colon cancer (classified as COAD), which was analyzed and found to have just over 40% of mutations in biogenesis genes. Four genes were found to be overmutated in the COAD dataset: SMAD4, RAN, KHSRP and SMAD2 [42]. The functions of these proteins and other accessory proteins in the biogenesis pathway are detailed in Table 1. For example, SMAD4 and SMAD2, when found in a complex, are involved in TGFβ/BMP signaling, and upon activation, they activate miRNA precursor transcription. SMAD2 also leads to an increase in DROSHA processing of primary miRNAs within the nucleus [39,42,43,44]. Moreover, RAN interacts with exportin5 and functions to export miRNAs out of the nucleus and into the cytoplasm [33]. KHSRP (KH-type splicing regulatory protein) promotes processing of a subset of miRNAs by binding to the terminal loop sequence [47].

These data are also supported by earlier publications, which specifically detail the role of dysregulated miRNA biogenesis genes in colon cancer. For example, Vychytilova-Faltejskova et al. measured the expressions of 19 genes involved in the miRNA biogenesis pathway in the tumor tissues of 239 CRC patients and matched adjacent tissues. Additionally, they examined CRC patients (*n* = 17) with liver metastasis using RT-PCR [68]. Significant overexpression of most analyzed genes was observed in tumor tissue as well as in liver metastasis. One exception was LIN28A/B, which functions to inhibit DICER1 processing by recruiting proteins that lead to the uridylation of precursor miRNAs [42,50,51,52]. High levels of DROSHA and TARBP2 were associated with shorter disease survival, while overexpression of XPO5, TNCR6A and DDX17 were detected in tissues of patients with shorter overall survival and poor prognosis. See Table 1 for more detailed information on how they function in miRNA biogenesis. This is in accordance with earlier works that showed that DICER1 was upregulated in colon cancer patients and was correlated with poorer clinical outcomes [69,70,71,72]. However, this finding conflicted with work by Faggad et al. that showed that downregulation of DICER1 was a prognostic factor in CRC and negative expression was correlated with shortened overall survival [73].

The above section highlights the evidence that genetic and epigenetic alterations are common in biogenesis genes in colon cancer. These alterations are theorized to lead to altered biogenesis that could lead to changes in isomiR generation during cancer progression.

## 6. Post-Transcriptional Modifications

### 6.1. Nucleotidyl Transferase Activity

There is additional evidence that miRNAs, like most RNAs, are post-transcriptionally regulated. Nucleotidyl transferase, usually with uridyl- or adenosyl-transferase activity, is able to add nucleotides at the 3′-terminus of RNAs acting as polymerases independently of the template. These additions are able to impact stability and affect the performance of miRNAs. They can also preferentially affect certain miRNAs. Indeed, multiple enzymes were found to have this activity, including MTPAP, PAPD4, PAPD5, ZCCHC6, ZCCHC11 and TUT1 [57]. For example, the oncogenic miR-21 has been shown to be regulated post-transcriptionally by PAPD5. These 3′ adenylation events target them for degradation by PARN. This adenylation has been shown to be decreased in tumors across multiple tissues in TCGA (*n* = 10,271 primary cancer). This demonstrates the importance of isomiR generation in physiological contexts, particularly in cancer [57]. However, not all post-transcriptional regulation leads to degradation. The opposite was seen for miR-122 that was adenylated by PAPD4 at its 3′ end, and this adenylation event increased the stability of the miRNA [58]. These two highlighted cases show the diverse ways post-transcriptional modifications of miRNAs can affect stability either positively or negatively.

Terminal nucleotidyl transferases (TENTs) including ZCCHC11 and ZCCH6 have been implicated in a miRNA turnover mechanism known as target-directed miRNA degradation (TDMD) [74,75,76,77]. TDMD refers to the mechanism by which extensive pairing between a miRNA and a target mRNA outside of the seed sequence leads to a conformational change in the AGO PAZ domain that causes the 3′ end of the miRNA to be exposed. This exposure leaves it open to enzymatic attack, known as tailing, which is followed by degradation. TENTs function by adding non-templated sequences to the 3′ end. This is hypothesized to be one of the contributing factors to why 3′ isomiRs are more abundantly detected than 5′ isomiRs. For example, AGO mutations that disrupt the 3′ end binding were found to cause 3′ end tailing with ZCCHC11 and ZCCH6 in HEK293T cells and subsequent degradation by the exonuclease DIS3L2 [75]. Those researchers complemented that study by utilizing TCGA datasets to identify cancer patients with mutations in AGO. They found more miRNA reads with 3′ modification in patients with mutations in the binding pocket of AGO than in patients with mutations in other regions [75]. TDMD and other miRNA turnover mechanisms have been thoroughly reviewed previously [78].

### 6.2. Adenosine Deaminase Acting on RNA

Another set of enzymes, known as adenosine deaminase acting on RNA (ADAR), recognizes double-stranded primary miRNAs and, by deamination of adenosines, causes adenosine-to-inosine sequence changes. This modification can affect processing by either DROSHA or DICER. Kawahara et al. showed that ADAR editing of primary miRNAs at two specific sites inhibited their processing by DICER and led to accumulation of precursor miR-151 in the cytoplasm [79]. Yang et al. demonstrated that editing of primary miR-142 by ADARs inhibited its processing by DROSHA and led to its degradation by Tudor staphylococcal nuclease-like (Tudor-SN), a ribonuclease specific to inosine containing primary transcripts [49]. Tudor-SN, which can degrade inosine-edited miRNAs, is also a component of the RISC [56]. Tudor-SN is one of the most upregulated genes in CRC. Tsuchiya et al. demonstrated Tudor-SN upregulation at the mRNA level in very early-stage samples of CRC and in cell lines [80]. This highlights another alteration in biogenesis pathway genes that could lead to alteration in miRNA generation.

Besides affecting processing, this editing by ADARs can also lead to changes in the target seed sequence if an ADAR targets a sequence within the seed sequence region. This change causes an inosine to be recognized by guanosine instead of thymine, which causes a new set of mRNA targets to be established [81]. A recent report has established the first comprehensive list of miRNA molecular A-to-I editing events in humans. They were found by large-scale mapping in human samples that identified 2711 potential precursor miRNA editing sites, which are close to 80% of all human primary miRNAs. This report even identified 367 editing sites in mature miRNAs in the human samples, and these editing events were enriched in the seed sequence [82]. Editing of the seed sequence can lead to a completely new set of target mRNAs.

Besides direct editing events, ADARs have also been shown to affect processing of miRNAs in hepatocellular carcinoma cell lines. A novel mechanism was discovered, by Qi et al., in which ADAR1 or ADAR2 interaction with DICER enhances processing from precursor miR-27a to mature miR-27a [83]. This discovery highlights the multifaceted role that a single biogenesis gene can play in miRNA-processing events, contributing to stability, degradation or even new target sets.

### 6.3. Single-Nucleotide Polymorphisms

Single-nucleotide polymorphisms (SNPs) also occur within miRNAs and have been studied for their association with cancer risk. The miR-27a gene, a known oncogene with elevated expression in CRC, has been investigated for its polymorphisms and association with CRC. One polymorphism, rs895819, is located within the loop of precursor miR-27a, which leads to the change of nucleotide A > G in the region hypothesized to affect the maturation process and to lead to increased expression of oncogenic miRNAs. A recent meta-analysis by Dai et al. analyzed several studies [84,85,86,87,88,89] and found that the miR-27a polymorphism was associated with an increased risk of CRC as well as breast cancer [90]. This SNP likely affects processing. Other SNPs in miRNAs have been found in the seed region and can cause large changes in the mRNA set they regulate. These are another source of variations in miRNA sequence heterogeneity.

## 7. Functional Importance of IsomiRs

The above sections have detailed the many ways that miRNAs can generate sequence variations at all stages of their maturation and how that can lead to isomiRs. These changes in miRNAs’ isoforms can affect stability as well as target selection. Variations at the 5′ end can cause a change or a shift in the seed sequence and have a large functional effect, conferring a new set of target genes [91]. In addition, changes in position 1 at the 5′ end can have a great effect on the half-life of a miRNA [92]. However, the majority of isomiRs are 3′ isomiRs. The 3′ terminus, although not involved in direct target binding, is believed to play a role in the stability of mRNA–miRNA interaction and also contribute to the specificity of binding [93,94]. These changes affect the loading of miRNAs into the miRISC. Analysis has shown that isomiRs also are distinctly associated with distinct AGO proteins [95]. Thus, miRNA sequence variations that lead to isomiRs have vast significance in the functional role they play in regulating gene expression. The study of miRNA isoforms is only a recent discovery, and this body of research will continue to grow as the field advances. The variability in isomiR sequences determines their stability and target selection, conferring broad translational control in normal tissues. Specific mutations in miRNAs may have clinical value as normal tissues become cancerous.

## 8. Specificity of IsomiRs in Cancer

In the previous section, we discussed the possible functional importance that isomiRs play. In this section, we will review studies that demonstrate the importance of isomiRs in cancer. Telonis at al. utilized TCGA dataset to determine if isomiRs could be used to distinguish between the 32 cancer types in TCGA. They built a classifier based on the presence or absence of isomiRs (90% sensitivity). They identified 7466 isomiRs that were generated from 807 arms and 767 miRNA loci. Only 48 were found in all of the datasets examined. They discovered isomiRs that were significantly present in one type of cancer and absent in others. They were also able to generate binarized miRNA-arm profiles that only determined if a miRNA arm would produce one or more isomiRs above a set threshold and whether this could achieve suitable performance for classification (83% sensitivity). For example, miR-215-5p is specific to COAD as are the isomiRs produced from miR-215-5p. When those researchers added in the expression data, the sensitivity of the classifier diminished in parallel with the expression profile of the archetype miRNA [96]. Another study sought to reduce the number of features to generate classifiers by utilizing a reliable set of cancer-associated 5′ isomiRs from TCGA isomiR expression data for multiclass tumor classification. This reduction in the number of isomiRs used for classification achieved an average sensitivity of 91.5%, with only 50 highly expressed 5′ isomiRs [97]. The above two studies underlie the possible prognostic role that isomiRs can have as research and technology continue to expand our understanding of and capabilities in cancer diagnosis. This is important, as isomiRs are believed to be tissue- and disease-state-specific.

## 9. Methods

In the above sections, a few studies were selected for review to give a broad overview of key concepts. In the section below, “isomiRs and Colorectal Cancer”, a comprehensive search was completed in PubMed using the search terms “isomiR” AND “colorectal cancer”, producing 13 results. Two were review articles and thus will not be discussed in detail below. Two other articles were excluded; one article was focused on lung cancer and another was a study performed in a porcine model of CRC and not human samples. Additionally, a search of “miR isoforms” AND “colorectal cancer” yielded 25 results. However, only one new article of relevance was found with these search terms. A majority of articles were thus excluded because they only discussed mRNA isoforms.

## 10. IsomiRs and Colorectal Cancer

Advances in next-generation sequencing have allowed for analysis of high-throughput sequencing data and have allowed for a great increase in the amount of data available at the isomiR level. This section will cover the data available for isomiRs in CRC.

### 10.1. Jiao et al., 2017

Utilizing TCGA database, Jiao et al. investigated the isoforms of miR-21-3p in CRC [98]. They found 10 isoforms in CRC patients, and these isoforms were found to mainly occur in proximal colonic adenocarcinoma and also microsatellite-instable (MSI) tumors. Higher expression of the isoform mir-21-3p 0|2 was a favorable marker for CRC patients in TCGA database. This naming convention denotes that the isoform is missing 2 nt downstream of the canonical terminus. They looked at several different isoforms and their effects on cellular migration in two colonic cell lines after transfection with the isoform mir-21-3p 0|2. The cells that were transfected with mir-21-3p 0|2 had the least number of migrating cells. They also showed that this isoform was uniquely predicted, using prediction software RNA22, to target three known oncogenes [98].

### 10.2. Dokanehiifard et al., 2017

Using bioinformatics analysis, Dokanehiifard et al., 2017, identified a novel miRNA within the tropomysorin receptor kinase C (TrkC) gene, named TrkC-miR2. They showed that this miRNA was significantly upregulated in human colonic tumors compared to matched normal samples. TrkC-miR2 was experimentally validated to target APC2, a WNT pathway component, and Dokanehiifard et al. established this miRNA as a novel regulator of the WNT pathway. During this experimental validation, they established two mature isomiR forms that vary in the last nucleotides at the 3′ end in the colonic human cell line SW480 as well as other non-colonic human cell lines. This highlights the evolving field of miRNA research, demonstrating the discovery of a novel miRNA with oncogenic potential as well as the discovery of two isoforms of this miRNA [99].

### 10.3. Wu et al., 2018

Wu et al. developed a comprehensive approach to sequence-oriented isomiR annotation (CASMIR) that enabled analysis of isomiR expression from small RNA-sequencing data [100]. They analyzed the sequencing data from 20 normal, 26 advanced adenoma and 25 formalin-fixed, paraffin-embedded (FFPE) CRC tissue samples. They found that isomiRs were the most abundant form of small RNA in CRC samples, contributing close to 70% of the total miRNA reads. Of these isomiRs, 3′ isomiRs were the most prevalent. IsomiRs with polymorphic changes in the 5′ seed regions were uncommon (<0.5%). This suggests that a majority of isomiRs may affect the same mRNA targets as do their canonical miRNAs. Wu et al. found that the levels of some isomiRs greatly exceeded those of their canonical miRNAs. For instance, miR-27a-3p (3′ deletion C), miR-125a-5p (3′deletion GA), miR-224-5p (3′ addition U) and miR-30e-5p (3′ addition CU) were all at least sevenfold higher than their canonical miRNAs. Wu et al. compared expression levels of miRNAs from CRC to normal, as well as advanced adenoma to normal (*n* = 2937 sequences analyzed). In comparison of advanced adenomas, 1250 sequences were downregulated and 144 sequences were upregulated. In comparison of CRC to normal cases, 631 miRNAs were downregulated and 131 were upregulated in CRC. When advanced adenomas and CRC were compared, 58 upregulated isomiR sequences were found in common. A majority of the upregulated isomiRs belonged to the miRNA families miR-17-92, miR-200 and miR-183, which have been reviewed extensively [19,100,101,102,103]. These families are well-studied and have been found to play a key role in cancer progression.

### 10.4. Mjelle et al., 2019

Mjelle et al. sequenced small RNAs from 48 early-stage I-II colon cancer samples and matched adjacent normal tissue pairs by utilizing Illumina high-throughput sequencing [104]. They identified 331 miRNAs as differentially expressed in their patient samples. In nearly 50% of these samples, the highest-expressed sequence were isomiRs. Of these isomiRs, Mjelle et al. identified 2451 differentially expressed isomiRs, which was close to 70% of all the isomiRs identified. Interestingly, these originated from only 343 miRNAs. Of note, miR-192-5p had 131 significant isomiRs, the most in the dataset. A recent review looked at the role of mir-192-5p in human disease [105]. Mjelle et al. also analyzed miRNA expression differences based on tumor location and microsatellite instability. They noted that miR-196b was upregulated in normal right colonic tissue but downregulated or unchanged in right and left tumor tissue. miR-615-3p and its isomiRs were the only miRNAs to show reliable expression differences between the left and right colon in both the normal and the disease state [104], demonstrating again that miRNAs are tissue- and disease-state-specific, including miRNA isoforms.

### 10.5. Nersisyan et al., 2021

A study by Nersisyan et al. sought to develop a regulatory network of specific genes involved in the extracellular matrix (ECM)-receptor interactor pathway in CRC [106]. They included transcription factors, miRNAs and miRNA isoforms with variable 5′ ends utilizing a tool they developed (miRGTF-net) that incorporates expression and database-level data to construct a regulatory network. They then utilized exhaustive search-based Cox model fitting to create miRNAs (including isomiRs)/gene signatures to predict overall survival in patients [94]. Sequencing at the mRNA and miRNA levels was obtained from TCGA COAD cohort, which included both normal and tumor samples and had clinical information provided. The network consisted of 522 nodes, 442 transcription factors and 27 miRNAs and 5′ isomiRs as well as 53 genes from the ECM-receptor pathway. Nersisyan et al. used a modified naming convention for annotating their 5′ isomiRs. A positive sign meant that an isomiR’s terminus was downstream from the canonical terminus (in the 5′→3′ direction); a negative sign indicated that an isomiR’s terminus was upstream of the canonical terminus. Thus, miR-200b-3p|+1 differed from the canonical miRNA because it was missing the first nucleotide at the 5′ end. Nersisyan et al. identified two miRNAs, miR-29b-3p and miR-32-5p, and one 5′ isomiR, miR-148a-3p|−1, that regulated the ECM set [94]. They found 39 significant miRNA–gene interactions, and 13 out of the 39 miRNA–target gene interactions were 5′ isomiRs, including miR-148a-3p|−1, miR-335-3p|−1, miR-30e-5p|+1, miR-92a-3p|+2, miR-203a-3p|+1, miR-200b-3p|+1, miR-194-5p|−1 and miR-142-3p|+1 [106]. This study highlights the important role isomiRs play in ECM receptor interaction: a pathway of importance in CRC progression.

### 10.6. Raigorodskaya et al., 2022

Raigorodskaya et al. looked at the colonic cancer cell line HT-29 and analyzed changes in isomiR expression after hypoxia exposure [107]. They experimentally mimicked hypoxia with two chemicals: cobalt II chloride and oxyquinoline. Both chemical treatments upregulated the canonical miRNAs miR-210-3p and miR-22-3p while downregulating let-7a-3p, miR-615-3p and miR-4521. The cells treated with cobalt II chloride uniquely downregulated four canonical miRNAs, miR-200a-5p, miR-425-5p, miR-32-5p and miR-1307-3p; four uniquely upregulated canonical miRNAs, miR-30b-5, miR-182-5p, miR-27a-5p and miR-215-5p; and two distinctively upregulated non-canonical isomiRs, miR-183-5p|+1 and miR-224-5p|+1, in which the +1 denotes that these isomiRs were missing the first nucleotide compared to the canonical miRNAs. The cells treated with oxyquinoline uniquely upregulated canonical miR-23a-3p only. Raigorodskaya et al. performed further analysis on the two upregulated non-canonical miRNAs, focusing on the targeting capabilities that were unique to them. When they analyzed these unique targets, they found that TCF4 decreased nearly tenfold and EGR1 decreased just over thirteenfold [107]. This study highlights the specificity of isomiRs in cellular states and how these cause great effects on downstream targets.

### 10.7. Lukosevicius et al., 2022

Lukosevicius et al. defined differential miRNA expression during the early transition from colon adenoma to carcinoma by performing small-RNA-sequence profiling of colon biopsy samples from healthy controls (HCs) and patients with CRC and adenomatous polyps (APs) [108]. They looked at the overall similarity of miRNA transcriptomes using multidimensional scaling analysis that allowed visualization of the similarities between the transcriptomes. This visualization and the subsequent analyses showed three clear clusters, with APs being an intermediate between the HC and CRC samples. Lukosevicius et al. then performed a differential analysis that identified 157 upregulated miRNAs and 51 downregulated miRNAs in the CRC samples compared to the HCs. They showed that, in comparing the APs with the HCs, there were 89 upregulated and 57 downregulated miRNAs. They also showed that between the CRC and the APs, there were 122 upregulated and 69 downregulated miRNAs. They then performed a three-way differential analysis that showed that 60 miRNAs were consistently upregulated in both the APs and the CRC compared to the HCs. The CRC samples had 67 exclusively elevated miRNAs, and the AP samples had 37 distinctive upregulated miRNAs. Lukosevicius et al. identified thousands of significantly dysregulated isomiRs, which were included in their supplementary data Table 2 [108]. Reviewing this data revealed that there were 1686 differentially expressed isomiRs in the Aps compared to the HCs, 2520 differentially expressed isomiRs in the CRC compared to the Aps, and 2343 differentially expressed isomiRs in the CRC compared to the HCs.

We performed additional analysis of the isomiRs in this dataset. First, we filtered Supplementary Table 2 from Lukosevicius et al. in order to examine the CRC compared to the healthy controls (HCs) only. We then filtered this further to select the top 55 expressed isomiRs based on the base mean level of expression (Appendix A). Out of these 55 isomiRs, only 24 were significantly dysregulated with adjusted *p*-values of less than 0.05 (Appendix A). Out of the 24 significant isomiRs, only four had positive log2 fold changes in comparison of the CRC to the HCs (Appendix A). One was an isomiR of miR-27a-3p, which will be discussed below. Three of the four significant upregulated isomiRs were miR-21-5p isomiRs (Appendix A). Compared to its isomiRs, the canonical miR-21-5 had the highest base mean expression and was similarly significantly increased in the CRC compared to the HCs (Appendix A). miR-21-5p is one of the most highly expressed and conserved miRNAs, and thus is widely studied [109]. miR-21-5p is highly expressed in CRC and considered an oncomiR. There have been many studies investigating the potential of miR-21-5p as a prognostic and diagnostic biomarker and a therapeutic target [110,111]. Out of the twenty isomiRs with negative fold changes (Appendix A), eleven were isomiRs of miR-192-5p (Appendix A), and the canonical one had the highest base mean level of expression and was also decreased in the CRC compared to the HCs. This is in line with Mjelle et al., who saw that miR-192-5p had the highest number of significant isomiRs and that canonical miR-192-5p decreased in CRC compared to HCs [103]. Moreover, miR-192-5p is considered to act as a tumor suppressor in CRC [112] and has been shown to be downregulated in CRC progression [113,114].

We chose to perform additional analysis on miR-27a-3p, the canonical miRNA that was within the top 40 of all of the normalized miRNAs over all samples in comparison groups; see Table 2. The canonical miRNA had a base mean of around 9610 in a normalized average count. It increased significantly between the APs and the HCs, between the CRC and the APs, and between the CRC and the HCs. Similar to what was reported by Wu et al., the most common mir-27a-3p isomiR was a 3′ deletion of the “C” nucleotide, although in this dataset, it was not more prevalent than the canonical miRNA. This is contrary to Wu et al.’s finding on the isoform miR-27a-3p, that the 3′ deletion was 18 times more abundant than in the canonical form [100]. All isoforms of miR-27a-3p that were significantly expressed were 3′ isomiRs, and in comparing the CRC with the HCs (by following the trend to the canonical miRNAs), all isoforms were significantly enriched in the CRC dataset. Although there were no changes in the seed sequence, these 3′ isomiR variants could affect stability as well as increase the competition for canonical targets. However, further research should be conducted to determine if there is functional relevance to these isomiRs.

## 11. Discussion and Conclusions

The discovery of miRNAs in the early 1990s has been followed by an explosion of research in the field. miRNAs have been shown to play a key role in human cancers, including all stages of CRC progression. One of the more recent discoveries is the presence of isoforms of miRNAs, which has been made possible by more sophisticated and sensitive sequencing technology. It is now well-established that generation of isomiRs is relatively common and they can be generated in several specific ways, although many unidentified sources are a likely possibly. As also noted by Zelli et al. in their recent review of isomiRs in cancer, a majority of the studies have largely involved profiling experiments of isomiRs and not on their possible functional role [115]. In our narrower review of isomiRs in CRC, a majority of the papers still largely focus on profiling experiments. As the field continues to mature, our understanding of the functional aspects of isomiRs will grow, in both their generation and functional significance. This will become of great importance, as miRNAs are being heavily researched as prognostic tools.

## Figures and Tables

**Figure 1 ncrna-09-00034-f001:**
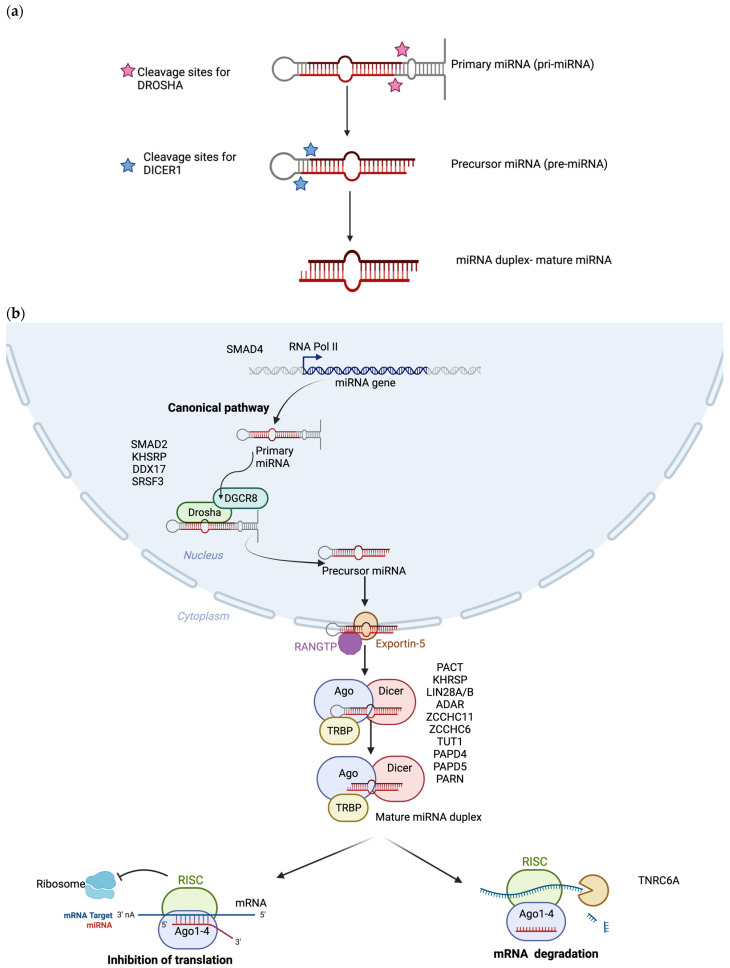
(**a**) Brief overview of the naming convention for miRNA maturation steps. Pink stars highlight where DROSHA performs the first cleavage within the nucleus. The blue stars highlight where DICER1 cleaves within the cytoplasm to produce a mature miRNA duplex. (**b**) RNA polymerase II transcribes miRNA genes as primary miRNAs, seen in Figure 1a. Primary miRNAs are recognized by DROSHA and DGCR8 and then cleaved to produce precursor miRNAs. The precursor miRNAs are exported to the cytoplasm by exportin5 and RAN-GTP. Then, in the cytoplasm, the precursor miRNAs are recognized by DICER1 and cleaved again to produce a short, double-stranded RNA duplex. The duplex is transferred to an AGO protein. One strand of the duplex is unwound and then degraded, leaving only the guide strand and forming the minimal miRNA-induced silencing complex. The guide strand is used to target mRNAs based on complementarity with the seed sequence. This complementarity determines if translation will be inhibited or if AGO-dependent splicing will occur. Names of accessory proteins, in black and to the side of the biogenesis step, are included in the figure and described in Table 1. Both images were created with BioRender.

**Figure 2 ncrna-09-00034-f002:**
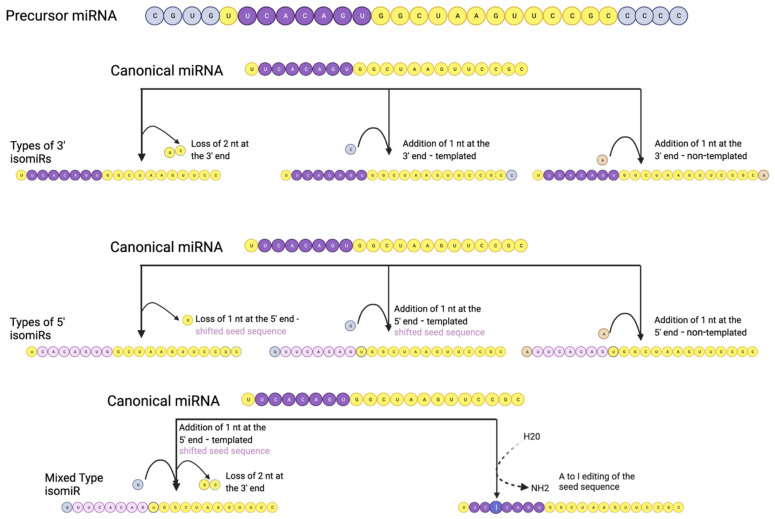
Visual representation of isomiR types, including 3′ isomiRs, 5′ isomiRs and mixed types. 3′ isomiRs are generated by additions or losses at the 3′ end. 5′ isomiRs are generated by losses or addition at the 5′ end. These changes can cause a shift in the seed sequence. Mixed-type isomiRs involve both 3′ and 5′ additions and/or losses. Additions are classified as templated or non-templated. Templated refers to nucleotides in the precursor sequence that are typically cleaved in the canonical miRNA but remain in the isomiR. The most commonly found isomiRs are 3′ isomiRs with non-templated additions. Image was created with Biorender.

**Table 2 ncrna-09-00034-t002:** Additional analysis of miR-27a-3p isomiRs available in [96].

**Adenomatous Polyps vs. Healthy Controls**			
**Gene**	**Base Mean**	**Log2 Fold Change**	**Adjusted *p*-Value**
miR-27a-3p	9614.9	0.343	0.006
miR-27a-3p.iso.t5:0.seed:0.t3:cgc.ad:GG.mm:0	64.7	0.681	0.004
miR-27a-3p.iso.t5:0.seed:0.t3:0.ad:AAT.mm:0	1.4	1.293	0.028
miR-27a-3p.iso.t5:0.seed:0.t3:c.ad:TAT.mm:0	3.6	0.824	0.032
miR-27a-3p.iso.t5:0.seed:0.t3:c.ad:0.mm:0	5347.8	0.336	0.038
miR-27a-3p.iso.t5:0.seed:0.t3:c.ad:AAC.mm:0	5.1	0.607	0.044
miR-27a-3p.iso.t5:0.seed:0.t3:c.ad:GAT.mm:0	0.9	1.259	0.045
miR-27a-3p.iso.t5:0.seed:0.t3:cgc.ad:0.mm:0	505.1	0.813	<0.001
miR-27a-3p.iso.t5:0.seed:0.t3:gc.ad:0.mm:0	1192.9	0.554	<0.001
**Colorectal Cancer vs. Adenomatous Polyps**			
**Gene**	**Base Mean**	**Log2 Fold Change**	**Adjusted *p*-Value**
hsa-miR-27a-3p	9614.9	0.343	0.006
miR-27a-3p.iso.t5:0.seed:0.t3:cgc.ad:0.mm:0	505.1	−0.675	<0.001
miR-27a-3p.iso.t5:0.seed:0.t3:cgc.ad:GG.mm:0	64.7	−0.671	0.010
miR-27a-3p.iso.t5:0.seed:0.t3:0.ad:A.mm:0	62.0	0.578	0.010
miR-27a-3p.iso.t5:0.seed:0.t3:C.ad:0.mm:0	7.2	0.668	0.017
miR-27a-3p.iso.t5:0.seed:0.t3:c.ad:TT.mm:0	119.3	0.701	0.017
miR-27a-3p.iso.t5:0.seed:0.t3:c.ad:TA.mm:0	5.3	0.705	0.025
miR-27a-3p.ref.t5:0.seed:0.t3:0.ad:0.mm:0	861.6	0.473	0.032
miR-27a-3p.iso.t5:0.seed:0.t3:c.ad:0.mm:0	5347.8	0.367	0.035
miR-27a-3p.iso.t5:0.seed:0.t3:gc.ad:TTT.mm:0	3.2	0.815	0.038
miR-27a-3p.iso.t5:0.seed:0.t3:gc.ad:A.mm:0	20.1	1.003	<0.001
**Colorectal Cancer vs. Healthy Controls**			
**Gene**	**Base Mean**	**Log2 Fold Change**	**Adjusted *p*-Value**
hsa-miR-27a-3p	9614.9	0.343	0.006
miR-27a-3p.iso.t5:0.seed:0.t3:gc.ad:0.mm:0	1192.9	0.793	<0.001
miR-27a-3p.iso.t5:0.seed:0.t3:c.ad:0.mm:0	5347.8	0.704	<0.001
miR-27a-3p.iso.t5:0.seed:0.t3:c.ad:T.mm:0	223.6	0.710	<0.001
miR-27a-3p.ref.t5:0.seed:0.t3:0.ad:0.mm:0	861.61	0.652	<0.001
miR-27a-3p.iso.t5:0.seed:0.t3:c.ad:TC.mm:0	2.0	1.190	0.002
miR-27a-3p.iso.t5:0.seed:0.t3:c.ad:AAC.mm:0	5.1	0.738	0.005
miR-27a-3p.iso.t5:0.seed:0.t3:0.ad:A.mm:0	62.0	0.568	0.007
miR-27a-3p.iso.t5:0.seed:0.t3:0.ad:AAT.mm:0	1.4	1.476	0.007
miR-27a-3p.iso.t5:0.seed:0.t3:gc.ad:A.mm:0	20.1	0.566	0.0218
miR-27a-3p.iso.t5:0.seed:0.t3:c.ad:ATT.mm:0	1.7	1.058	0.025
miR-27a-3p.iso.t5:0.seed:0.t3:c.ad:AC.mm:0	1.7	1.004	0.029
miR-27a-3p.iso.t5:0.seed:0.t3:0.ad:G.mm:0	17.6	0.388	0.047
miR-27a-3p.iso.t5:0.seed:0.t3:c.ad:TCT.mm:0	1.7	0.851	0.048

Results of miRNA differential expression analysis among CRC, AP and HC tissues (NGS data). Column annotations: Gene—miRNA name; Base Mean—the average of the normalized counts taken over all samples in comparison; log2 Fold Change—log2 fold change between the groups; Adjusted *p*-Value—Benjamini–Hochberg adjusted *p*-value; comparison—compared group. Footnote adapted from [108]. The type of miRNA is an isoform if the sequence is followed by .iso. The t5 tag indicates variations at the 5′ end; a “0” indicates no variations. The seed tag indicates changes that happen between nucleotides 2 and 8. The t3 tag indicates variations at the 3′ position. The naming contains two words, “direction-nucleotides”, where the direction can be lowercase NT (upstream of the 3′ reference position) or uppercase NT (downstream of the 3′ reference position). After “direction” follows the nucleotide/s that are added (for downstream changes) or deleted (for upstream changes). ad tag: indicates nucleotide additions at the 3′ position. The naming contains two words, “direction-nucleotides”, where the direction is uppercase NT (upstream of the 5′ reference position). “0” indicates no variation, meaning the 3′ position has no additions. After “direction” follows the nucleotide/s that are added. mm tag: indicates nucleotide substitutions along the sequences. Annotation information can be found at https://bioc.ism.ac.jp/packages/3.3/bioc/vignettes/isomiRs/inst/doc/isomiRs-intro.pdf, accessed on 19 March 2023.

## Data Availability

Data is available by request from corresponding author.

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
