# Peer review of "A Review of IsomiRs in Colorectal Cancer"

_ncrna, 2023, doi:10.3390/ncrna9030034_

Round 1

Reviewer 1 Report

I would elaborate more on miR-451 for two reasons, 1) it is the only known miRNA whose processing bypasses Dicer and but rely on slicer activity of Ago2, 2) it is highly abundant in eryothrocytes, played a role in RBC maturation, and has implication as a RBC biomarker, for example for detecting fecal occult blood.

Would be useful to mention the idea of small RNA (17-25 bp) resulting from RNA degradome and how this species is different from miRNA (namely miRNA is very strictly defined by its biogenesis)

miRNA nomenclature needs to be more consistent throughout the article, sometime hsa- is used sometimes it's not. And for Nersisyan et al.'s 2021 paper the nomenclature needs to be elaborated as there is currently no standard methods in naming isomiRs.

Author Response

Comment 1.1: I would elaborate more on miR-451 for two reasons, 1) it is the only known miRNA whose processing bypasses Dicer and but rely on slicer activity of Ago2, 2) it is highly abundant in eryothrocytes, played a role in RBC maturation, and has implication as a RBC biomarker, for example for detecting fecal occult blood.

  • Thank you for this suggestion. We have elaborated more on miR-451 in our revised manuscript.

Comment 1.2: Would be useful to mention the idea of small RNA (17-25 bp) resulting from RNA degradome and how this species is different from miRNA (namely miRNA is very strictly defined by its biogenesis)

  • Thank you for this clarifying suggestion. We have added this concept to our revised manuscript.

Comment 1.3: miRNA nomenclature needs to be more consistent throughout the article, sometime hsa- is used sometimes it's not.

  • Thank you for pointing this out we have removed hsa from document.

Comment 1.4: And for Nersisyan et al.'s 2021 paper the nomenclature needs to be elaborated as there is currently no standard methods in naming isomiRs.

  • We apologize for the lack of clarity and have updated the manuscript to improve our description.

Reviewer 2 Report

Lausten and Boman introduce and descrive the state of the art on isomiRs in colorectal cancer.

The authors  provide a clear and comprehensive overview of the subject for readers unfamiliar with the topic.

The review describes miRNA biogenesis and role in colorectal cancer, and focuses on the discovery e description of miRNA isoforms, the so-called isomiRs. The authors discuss the studies that were published on this topic, to provide an overview about isomiR role in colorectal cancer. The review is well-written. I found a few inaccuracies that should be corrected before the publication: 

  • The term isomeric miR did not refer to miRNA variant/isoform, the so-called isomiR (see publication with following PMID 29657271). Please replace with the correct wording. 
  • The sentence in lines 396-398 is not clear: “They used an isomiR naming convention in which +1 is the number that represents the offset at the 5’ end, whereby +1 means the isomiR is missing the first nucleotide at the 5’end”. Please revise the sentence. 

Author Response

Comment 2.1: The term isomeric miR did not refer to miRNA variant/isoform, the so-called isomiR (see publication with following PMID 29657271). Please replace with the correct wording. 

  • Thank you for providing that article! We have updated the language accordingly.

Comment 2.2: The sentence in lines 396-398 is not clear: “They used an isomiR naming convention in which +1 is the number that represents the offset at the 5’ end, whereby +1 means the isomiR is missing the first nucleotide at the 5’end”. Please revise the sentence. 

  • We apologize for the lack of clarity and have updated the manuscript to improve our description of the nomenclature.

Reviewer 3 Report

This review by Lausten and Boman describes a comprehensive picture of isomiRs in colorectal cancer. Remarkably, the authors present an extensive review of the current literature in vast areas such as the miRNA biogenesis, functions and impact of isomiRs. Given the complexity and the vast literature available, this is an excellent review.

Major comments:

  • Some isomiRs differentially expressed in tumours may reflect changes in expression of RNA-binding proteins such as SRSF3 or other factors such as those mentioned in Table 1. Nevertheless, changes in the expression of these factors can also have effects on other parts of the RNA metabolism such as splicing, RNA decay... Maybe a paragraph should be introduced to discuss these potential confounding effects.

  • On the other hand, any function related to isomiRs is expected to be subsequent to their relative abundance to the canonical sequences and to their absolute abundance. The manuscript may benefit from providing insights on which isomiRs dysregulated in colorectal cancer have the highest CPMs, or highest relative percentages to the canonical product.

  • It would be very illustrative, in Figure 2 to present some numbers of relative abundance of the different isomiR classes. I think there is general consensus that 3’ end tailing (TENT activity) is highly predominant, followed by 5’ isomiRs (Drosha and Dicer miscleavage). By contrast, ADAR activity on mature miRNA or SNPs affecting mature miRNA is rather reduced. Also, specific examples of miRNAs where one of the isomiR classes is highly represented would be helpful.

  • It would be important to explain that tailing of mature miRNAs is dependent on TENTs by also on the availability of miRNAs 3’ end. Dislocation from the PAZ AGO proteins by pairing the miRNA sequence with RNAs with high complementarity has been proposed as a major trigger for TENT activity (PMID: 31353209, 32488030, 34819352).

Author Response

Comment 3.1: Some isomiRs differentially expressed in tumours may reflect changes in expression of RNA-binding proteins such as SRSF3 or other factors such as those mentioned in Table 1. Nevertheless, changes in the expression of these factors can also have effects on other parts of the RNA metabolism such as splicing, RNA decay. Maybe a paragraph should be introduced to discuss these potential confounding effects.

  • Thank you for this helpful suggestion to improve our manuscript. We have added discussion of the concepts stated above to our revised manuscript.

Comment 3.2: On the other hand, any function related to isomiRs is expected to be subsequent to their relative abundance to the canonical sequences and to their absolute abundance. The manuscript may benefit from providing insights on which isomiRs dysregulated in colorectal cancer have the highest CPMs, or highest relative percentages to the canonical product.

  • Thank you for this suggestion. We agree this would be helpful. We have included additional discussion of this in our updated manuscript.

 Comment 3.3: It would be very illustrative, in Figure 2 to present some numbers of relative abundance of the different isomiR classes. I think there is general consensus that 3’ end tailing (TENT activity) is highly predominant, followed by 5’ isomiRs (Drosha and Dicer miscleavage). By contrast, ADAR activity on mature miRNA or SNPs affecting mature miRNA is rather reduced. Also, specific examples of miRNAs where one of the isomiR classes is highly represented would be helpful.

  • Thank you for this comment we agree. We have added a few simple sentences in a couple parts of our revised manuscript to emphasis the importance of the relative abundance of the isomiR classes. However further analysis on the relative abundance is part of our on-going research and will be presented in a separate study.

Comment 3.4: It would be important to explain that tailing of mature miRNAs is dependent on TENTs by also on the availability of miRNAs 3’ end. Dislocation from the PAZ AGO proteins by pairing the miRNA sequence with RNAs with high complementarity has been proposed as a major trigger for TENT activity (PMID: 31353209, 32488030, 34819352).

  • Thank you for this comment. We have updated our manuscript to incorporate this information.